# Fixed-Bed and Mobile-Bed Resistance of Channels with Steep Gradients in Mountainous Areas

**Ji Hou [1], Chunze Zhang [1], Dan Wang [1,2], Feng Li [1], Zijian Yu [2] and Qin Zhou [1,2,*]**

[1] Southwest Water Transport Engineering Research Institute, Chongqing Jiaotong University, Chongqing 400016, China; xks-houji@foxmail.com (J.H.); zhangchunze@whu.edu.cn (C.Z.); wangdan_cherry@126.com (D.W.); sculifeng@163.com (F.L.)

[2] State Key Laboratory of Hydraulics and Mountain River Engineering, Sichuan University, Chengdu 610065, China; isrs2019@yeah.net

* Correspondence: paulacton@126.com

**Abstract:** Flood discharge and sediment transport are closely linked to channel resistance in steep mountain streams. Previous research has mainly focused on the resistance of fixed-bed channels with steep gradients and mobile-bed channels in alluvial rivers. The present study performs an experiment and establishes a calculation method for the fixed-bed resistance of mountain channels. The basic expression of the mobile-bed resistance of steep mountain channels is derived by determining the controlling factors of the bed load movement on the riverbed resistance. The proposed formula can accurately predict the variation of the bed load resistance. The results of the present research improve the understanding of fluid dynamics and sediment transport in steep mountain channels.

**Keywords:** channels with steep gradient; mountainous river; fixed bed resistance; mobile bed resistance

## 1. Summary

The bed resistance of rivers is an important basis for flood control and engineering design. On the other hand, it is one of the basic challenges in hydraulics and river dynamics. Research on the resistance of mountainous rivers is even more challenging [1]. The most striking feature of rivers in the southwestern mountainous region of China is the narrow valleys with steep slopes. Bed material is mostly comprised of pebbles and gravel, which are coarse particles with wide gradation. Due to the unique topographical condition and bed surface composition, the flow characteristics are a low water depth, large flow velocity, large Froude number, and variable flow patterns. In order to understand the bed load transfer of mountainous rivers, explore the downward flow, and study the sediment transport law, it is necessary to understand the resistance of mountainous rivers and the important parameter of hydraulic factors.

In mountainous rivers, when the sediments are still or transported with low-intensity, the presence of sediment particles on the riverbed surface and their impacts on the water flow are regarded as the fixed bed resistance of the steep gradient channel. The resistance characteristics of the mountainous river, composed of coarse bed material, are quite different from those of alluvial rivers with fine bed particles. One of the important influencing factors is the relative roughness of the resistance of mountainous rivers [2–5]. Based on the relative roughness, researchers have divided the roughness degree into three levels. The division criteria of two studies are slightly different [5,6]. Other researchers have established resistance expressions [3,4,7–9]. In channels with a steep gradient in a mountainous area, the water flow is shallow, the bed material is coarse, and the intermediate scale roughness is more common, while large-scale roughness mostly occurs in a ladder-deep pool or in a pebble shoal.

The flow resistance of a steep riprap, located in constructed channels, was investigated by previous researchers [10,11]. Flume experiments or field data show that the resistance of the steep gradient channels and alluvial rivers is related to the size, shape, and gradation of sediment particles and the geometric shape of rivers. The characteristics of the resistance of two conditions are different and the differences in the quantitative studies cannot be ignored. Most of the studies used the measured data of a certain river or river section to fit the expression [3,12–15]. Large error values existed in formulas due to the different choice of roughness size and the differences of the data. The present work continues to explore the fixed-bed resistance characteristics of steep gradient channels through a flume experiment. Moreover, it intends to analyze the fixed bed resistance of the non-uniform bed surface gradient, based on a large number of measured data. It is expected that the recommended formula can control the error within a good range.

In order to consider the sediment movement impact of the riverbed on the water flow structure, it is necessary to study the mobile bed resistance of the mountainous rivers. The mobile bed resistance for alluvial rivers is the resistance of sand waves and develops at various stages. However, the large amount of coarse grain mountainous pebbles in the riverbed makes it difficult to form sand waves. Therefore, the resistance of the mobile bed mainly comes from the resistance caused by the consumed flow energy to maintain the pebble bed movement. At present, researchers focus on the resistance of steep gradient channels with sediment-laden flow to derive semi-empirical and empirical formula, mostly based on flume experiments and field data. The formulas are mainly obtained through three methods.

First, a large amount of experimental and measured data is used to fit a logarithmic formula, similar to the one for the fixed bed resistance. For steep channels with high relative flow submersion, researchers [16,17] found that the flow resistance has a correlation with relative submersion and it was found that the correlation has a logarithmic form. The corresponding hydraulic roughness as a single characteristic roughness length scale is frequently determined based on the equivalent grain size of the riverbed. Researchers conducted a flume experiment study on the resistance law and derived the expression, as follows [15]:

$$\sqrt{\frac{8}{f}} = 0.67 + 3.2\ln(\frac{R}{D}) \tag{1}$$

where $f$, $R$, and $D$ denote the friction factor, hydraulic radius, and grain size, respectively.

The research [15] shows that when there is a bed load movement in the water flow, the resistance is 2.5 times higher than the fixed bed resistance. The river resistance is constant and the resistance value is between that for a stationary condition and initiation sediment motion. The resistance of sediment-laden flow increases as the sediment transport rate increases. There is a significant defect in such expressions for the resistance. The variables in the formula are the hydraulic radius and the sediment particle size, while the impact of the sediment transport rate is ignored.

The second method assumes that the resistance of the sediment flow is the superposition of the fixed bed resistance with the same roughness and the resistance, caused by the bed load movement. Under this circumstance, researchers mainly studied the law of resistance change of the bed load movement [18]. The law of the resistance change in the bed movement is

$$f = f_c + f_b \tag{2}$$

where $f_c$ is the fixed bed resistance. $f_b$ is the resistance, caused by bed load. There is a nonlinear correlation between the resistance expression, caused by the bed load and the sediment concentration:

$$f_b = 0.00025 C_v^{1.23} \tag{3}$$

where $C_v$ is the dimensional mass concentration. It is the ratio of the volumetric flow of sediment ($Q_s$) to the discharge flow which carries the sediment *(Q)*:

$$C_v = Q_s/Q \tag{4}$$

Gao [18] believed that the transport concentration, relative roughness, and dimensionless particle size are the resistance affecting factors of the bed. They used the flume test and experiment data to establish an expression for the resistance of the bed load movement, based on dimensionless analysis [14]:

$$f_b = 0.048 C_v^{0.25} d_*^{0.5} \left(\frac{h}{D}\right)^{-0.75} \tag{5}$$

where $d_* = \left(\frac{r_s-r}{r}\frac{g}{n^2}\right)^{1/3} D_{50}$. Moreover, $\rho_s$, $\rho$, and $v$ are the sediment density, fluid density, and the coefficient of the fluid motion, respectively.

Based on the research results of Gao and Abrahams [18], a series of experiments are conducted to study the sediment resistance of the slope flow. The Froude number ($F_r$) of the water flow, as an effective factor in the flow resistance, is considered in accordance with the study of Gao and Abrahams [18] to establish an expression for calculating the resistance of the slope flow.

$$f_b = 282.5 C_v^{0.579} d_*^{-0.8} \left(\frac{h}{D}\right)^{0.25} F_r^{-3.539} S^{1.195} \tag{6}$$

where *S* is the gradient of the water surface.

The third method is used to compare the resistance coefficient ratios of the mobile and fixed beds and fit the expression of the ratio and influencing factors. Researchers have conducted a mobile bed experiment [14]. They considered the dimensionless particle size and sediment transport concentration as the influence factors and derived the following expressions for the flow resistance under the condition of bed load movement:

$$\frac{f}{f_c} = \left(30.1\sqrt{d_*}C_v + 1\right)^{0.92} \tag{7}$$

$$\frac{h}{h_c} = \left(\frac{f}{f_c}\right)^{1/3} \tag{8}$$

where $h$ and $h_c$ are the water depth during the bed load movement and the clear water depth under, respectively.

However, few researchers have focused on the flow resistance of mountainous channels. The general studies of moving bed resistance focus on the sand wave resistance of alluvial rivers at various stages. In the past, most of the experiments in the field of large-scale river channel mobile bed resistance were performed to study the resistance characteristics of water flow with sediment through the destruction of the bed surface. There are few studies focused on the influence of the sediment motion impact on river resistance of the stable bed surface. The present work carries out the flume experiment to analyze the variation of the resistance of the mobile bed in steep gradient channels.

## 2. Experiment Setup

The resistance of the bed with a uniform sediment experiment of steep gradient channels was carried out at the State Key Laboratory of Hydraulics and Mountain River Development and Protection, Sichuan University. The length, width, and height of the flume were 12.0 m, 0.4 m, and 0.4 m, respectively. Moreover, the slope was set to 10%, and was set in accordance with the mean gradients in mountain streams of southwest China. Figure 1 presents the configuration of the flume.

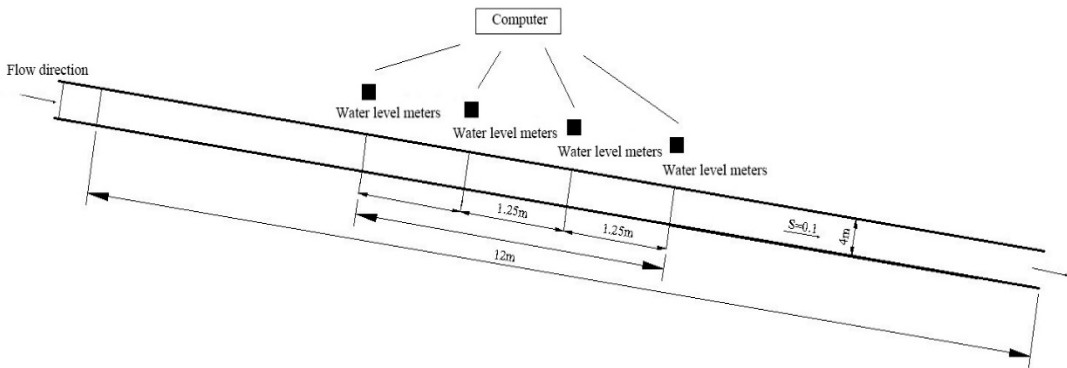

**Figure 1.** Schematic of the flume layout.

The sides of the flume were made up of 1 cm thick tempered glass, which were fixed to the aluminum alloy frame structure. The head of the flume had water control valves and static pools. Water flowed through the rectangular weir into the static pool. Then, the flow passed through the two-stage energy dissipation grille before it flowed into the flume. The flume end was provided with a flat gate and a sand grafting device to control the water level by adjusting the opening of the gate so that an approximately uniform flow was obtained. It should be indicated that the flow was considered to be uniform when the gradient of the water surface was close to the slope of the bed surface.

An automatic water level meter was mounted horizontally on the aluminum alloy rack at the top of the flume. In this experiment, the length of the test section was 3.75 m. Four automatic water level gauges were arranged on the test section, where each water level meter worked independently. The test data was recorded by connecting to the same computer. In this fixed bed with uniform sediment experiment, natural pebbles were laid in the base of whole flume, where the layer thickness was 3 cm. In order to ensure the reliability, representativeness, and comparability of the data, the bed material was made of two kinds of uniform sediments with particle sizes of 9.2 mm and 12.3 mm.

In the experiment of sediment laden resistance law in the steep gradient channel, the uniform sediment (non-uniform coefficient less than 1.3) with four particle sizes (3.8 mm, 6.7 mm, 9.2 mm, and 12.3 mm) was selected as the sediment entry channel. A funnel was arranged in the inlet section to control the sediment supplement rate through the opening of the funnel. Devices were installed at the end of the flume to measure the sediment transport rate. Four automatic water level meters simultaneously recorded the water depth variation of each section.

In this paper, the bed surface was assumed to be stable. Therefore, the pebble on the bed surface was fixed in the flume with a little cement slurry to simulate the stable bed surface. The relative discharge error was less than 3% when the flow passed through the rectangle thin-wall weir.

The first step of the sediment supplement was to put the prepared sediment into the flume through the sanding funnel. Then, different sediment transport rates were obtained through the bottom opening degree. The water level meter recorded the water level variations before and after the sanding funnel and recorded the bed surface elevation in advance. Based on the data from the spot measurement, a relatively accurate water depth variation could be obtained by averaging. The water level was controlled by the downstream baffle to make the water flow sufficiently stable with uniform turbulence. The added sediments neither accumulated nor exchanged with the sediment on the bed surface and were finally carried downstream by the flow.

## 3. Data Description and Method

### 3.1. Experimental Study on the Fixed Bed with Uniform Sediment Resistance of Steep Gradient Channnel

The coefficients of the river resistance are mainly expressed by the Chezy coefficient ($C$), the Darcy-Weisbach coefficient ($f$), and the Manning roughness coefficient ($n$). There is a mutual transformation correlation between these coefficients:

$$\frac{C}{\sqrt{g}} = \frac{R^{1/6}}{\sqrt{g}}\frac{1}{n} = \sqrt{\frac{8}{f}} = \frac{U}{U_*} \tag{9}$$

where $U$ is the average flow velocity.

In order to adjust the test setup to large specific slope setting, the flow rate is set to higher than 1 m/s and the variation of the water depth is limited to 1.63 cm to 6.18 cm. Especially, the drag coefficient presents the effect of several phenomena, such as bed interaction and non-uniform velocity distribution [19]. Since the above expression of the drag coefficient is equal to the ratio of the average flow velocity to the friction velocity, the ratio can be obtained by integrating the flow velocity distribution. Therefore, the study of the resistance is actually a study of the flow velocity distribution.

According to the sidewall roughness and flow conditions, the fixed bed resistance can be divided into the resistance of smooth, transition, and rough regions [20]:

$k_s/\delta > 10$; Rough region
$k_s/\delta < 0.25$; Smooth region
$0.25 < k_s/\delta < 10$; Transition region

Aguirre-Pe et al. [13] and Graf et al. [6] pointed out that the flow velocity above the bed surface in the vertical direction is basically in line with the logarithmic distribution, as follows [6,13]:

$$\frac{u}{u_*} = \frac{1}{\kappa}\ln\frac{y}{k_s} + B \tag{10}$$

where $u_*$, $\kappa$, and $k_s$ are the friction velocity, Karman constant (value is 0.4), and bed roughness size, respectively. It should be indicated that the Karman constant is set to $\kappa = 0.4$. $B$ is a constant. Montes [21] and Aguirre-Pe et al. [13] found that when the bed material composition in the steep gradient channel is uniform, the flow velocity on the surface of the river bed is constant within a certain range [11,21]. This range can be expressed as: $z_0 = \alpha k_s$, where $\alpha$ is a constant. The average flow velocity of the uniform sediment (see Figure 2) bed surface is obtained as follows:

$$U = \frac{u_0 z_0}{h} + \int_{z_0}^{h} u\,dy \tag{11}$$

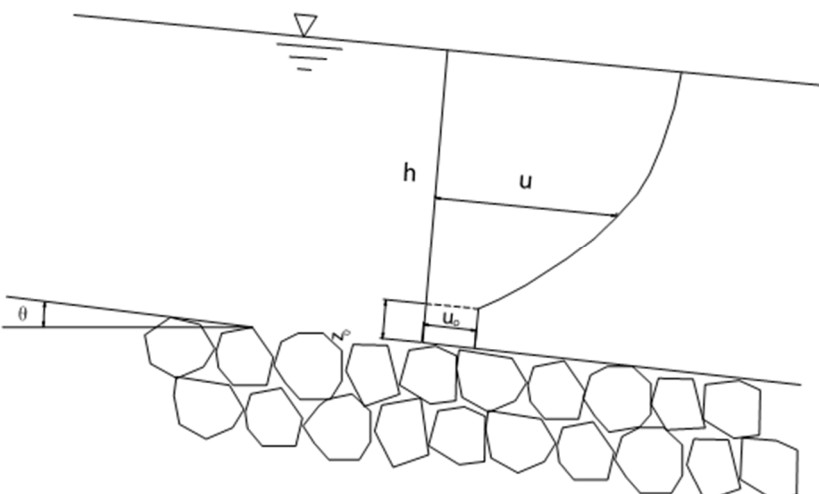

**Figure 2.** Schematic diagram of the flow velocity distribution for uniform sediment bed.

According to Equations (10) and (11):

$$\sqrt{\frac{8}{f}} = \frac{1}{k} \ln \frac{h}{k_s} + B - \frac{1}{k} + \frac{z_0}{kh} + \frac{z_0}{h} \left( \frac{u_0}{u_*} - \frac{1}{k} \ln \frac{z_0}{k_s} - B \right) \tag{12}$$

Since $y = z_0$ and $u = u_0$, Equation (12) can be re-written as the following:

$$\sqrt{\frac{8}{f}} = \frac{1}{\kappa} \ln \frac{h}{k_s} + B - \frac{1}{\kappa} + \frac{z_0}{\kappa h} \tag{13}$$

In the fixed bed resistance of the uniform sediment river bed, $k_s$ is the particle size of the sediment particles $D$. Moreover, according to the data of steep gradient channels, $z_0 = 0.2D$ [12,13,22,23]. Based on the experimental data in this paper, combined with the data of Song and Recking et al. (see Table 1) [15,24], the best value of parameter $B$ is 6.5.

**Table 1.** Data range of former researchers.

| Type | Number of Test | S | D (cm) | h (cm) |
|---|---|---|---|---|
| Song | 16 | 0.5–1.5 | 1.23 | 7.7–20.4 |
| Recking | 62 | 1.0–9.0 | 0.23–0.9 | 1.07–8.33 |
| Paper's experiment | 18 | 10.0 | 0.92, 1.23 | 1.63–6.18 |

Graf et al. [6] deduced the following expression for the fixed bed resistance:

$$\sqrt{\frac{8}{f}} = \frac{u}{u_*} = \frac{1}{\kappa} \ln \frac{R}{k_s} + B_r \tag{14}$$

where $B_r$ is a constant as the following:

$$\begin{cases} R/k_S > 20, & B_r = 6.5 \\ 4 < R/k_S < 20, & 3.25 < B_r < 6.5 \\ R/k_S < 4, & B_r = 3.25 \end{cases} \tag{15}$$

Figure 3 compares the data of Song and Recking [15,24] and the experimental results of the present study, obtained from Equations (12) and (14). Table 2 shows the hydraulic conditions and results of each group.

The results show that all these points fall within the range of expressions proposed by Graf [6]. Equation (12) can better reflect the fixed bed resistance of the uniform sediment in the steep gradient channels.

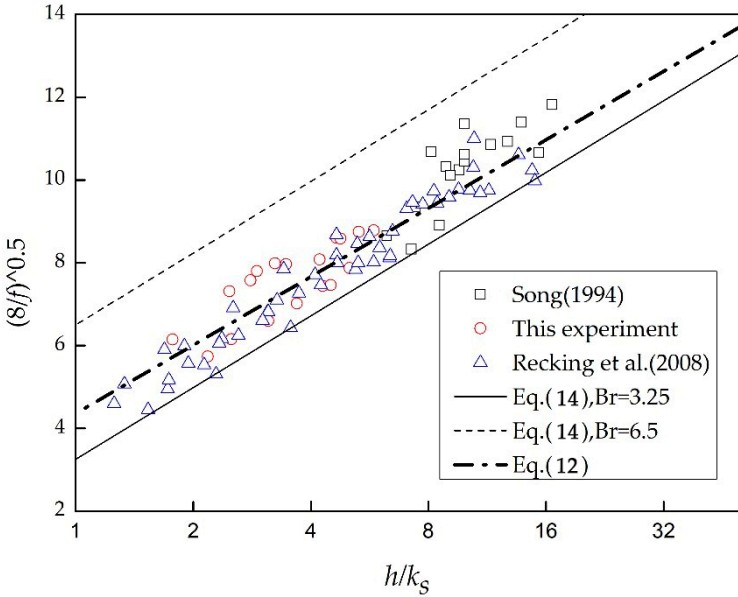

**Figure 3.** Comparison of the test data with the calculated values from Equations (12) and (14).

**Table 2.** Uniform sediment bed resistance test data of the steep gradient channel.

| No. | Discharge $Q$ (m³/s) | Particle Size $D$ (cm) | Gradient Ratio | Water Depth $h$ (cm) | Flow Velocity $U$ (m/s) | $R$ (cm) | $R/d$ | $(8/f)^{0.5}$ |
|---|---|---|---|---|---|---|---|---|
| 1 | 0.0051 | 0.92 | 0.1 | 1.63 | 0.777 | 1.507 | 1.638 | 6.146 |
| 2 | 0.0100 | 0.92 | 0.1 | 2.28 | 1.093 | 2.047 | 2.225 | 7.310 |
| 3 | 0.0124 | 0.92 | 0.1 | 2.58 | 1.205 | 2.285 | 2.484 | 7.577 |
| 4 | 0.0135 | 0.92 | 0.1 | 2.68 | 1.264 | 2.363 | 2.569 | 7.798 |
| 5 | 0.0163 | 0.92 | 0.1 | 2.98 | 1.365 | 2.594 | 2.819 | 7.985 |
| 6 | 0.0179 | 0.92 | 0.1 | 3.18 | 1.407 | 2.744 | 2.982 | 7.969 |
| 7 | 0.0244 | 0.92 | 0.1 | 3.88 | 1.575 | 3.250 | 3.532 | 8.078 |
| 8 | 0.0302 | 0.92 | 0.1 | 4.28 | 1.762 | 3.526 | 3.832 | 8.604 |
| 9 | 0.0312 | 0.92 | 0.1 | 4.38 | 1.779 | 3.593 | 3.906 | 8.585 |
| 10 | 0.0373 | 0.92 | 0.1 | 4.88 | 1.913 | 3.923 | 4.264 | 8.748 |
| 11 | 0.0428 | 0.92 | 0.1 | 5.33 | 2.006 | 4.208 | 4.574 | 8.779 |
| 12 | 0.0100 | 1.23 | 0.1 | 2.68 | 0.930 | 2.363 | 1.921 | 5.736 |
| 13 | 0.0132 | 1.23 | 0.1 | 3.08 | 1.069 | 2.669 | 2.170 | 6.154 |
| 14 | 0.0196 | 1.23 | 0.1 | 3.83 | 1.278 | 3.214 | 2.613 | 6.595 |
| 15 | 0.0268 | 1.23 | 0.1 | 4.53 | 1.478 | 3.693 | 3.003 | 7.015 |
| 16 | 0.0358 | 1.23 | 0.1 | 5.28 | 1.693 | 4.177 | 3.396 | 7.444 |
| 17 | 0.0384 | 1.23 | 0.1 | 5.53 | 1.736 | 4.332 | 3.522 | 7.459 |
| 18 | 0.0479 | 1.23 | 0.1 | 6.18 | 1.937 | 4.721 | 3.838 | 7.870 |

### 3.2. Investigation of the Resistance Characteristics of the Non-Uniform Sediment River Bed

In natural mountainous rivers, the riverbed surface is mostly composed of uneven coarse and fine particles. It is of great importance to study the resistance of the non-uniform bed surface of the steep gradient rivers and the effects of concealment and exposure between the coarse and fine particles in the water flow [3,8,12,25].

For the fixed bed resistance of the non-uniform bed material in the cobble channel of the mountainous area, researchers have analyzed a huge amount of measured field data and flume experiments to obtain an applicable expression for the resistance in a certain river section or channel. For the fixed bed resistance in mountainous rivers, it is difficult to introduce a universally applicable empirical expression through the measured data or the flume experiment data. Therefore, the estimation error is 25–30%, which is acceptable for practical applications [3].

By collecting a large amount of field measured and experimental data, it is found that the error value of the expression proposed by Hey, Griffiths and Graf [8,12,25] for the large-scale pebble channel is greater than 50%.

The basic expressions of resistance of the mountainous rivers are basically consistent. However, the different data selected by the researchers in the process of derivation and verification caused minor differences in the $k_s$ value and coefficients. According to the analysis discussed above, the uniform expression of the fixed bed resistance can be written as

$$\sqrt{\frac{8}{f}} = A \log\left(\frac{h}{k_s}\right) + L \tag{16}$$

where $A$ and $L$ are constants. Different scholars have proposed different values for A. However, the proposed values mostly vary from 5.6 to 5.75. The average value (i.e., $A = 5.70$) is considered in this study. At present, most studies choose $k_s = \alpha D_{50}$. Kikkawa et al [26] believed that the roughness size $k_s$ is independent of the particle size. However, the former value reflects the protruding height and geometry of the sand wave [26]. It is assumed that in mountainous rivers, the probability of sand waves occurrence is extremely small so the influence of sand waves is neglected in the present study. When the non-uniformity coefficient of the bed load is large and the gradation is wide, $D_{84}$ is selected for the roughness size. For the case where the non-uniformity coefficient is small and the gradation is narrow, $D_{50}$ is selected for the roughness size.

It is assumed that the fixed bed resistance is mainly affected by the bed material area on the riverbed plane, along the water depth. Therefore, the representative particle size $D_j$ is initially determined, and $k_s$ is calculated through $k_s = \alpha D_j$. Based on the above assumptions, the proposed calculation method of the present study is

$$\alpha = \frac{D_{50}}{D_j} \frac{\sum\limits_{i}^{n} p_i D_i{}^4}{D_{50}{}^4} \tag{17}$$

where $D_i$ can be divided into the following cases (see Figure 4):

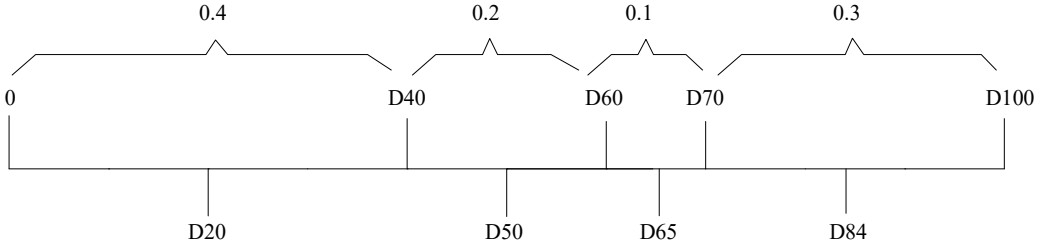

**Figure 4.** The distribution of $D_i$.

According to the choices of $D_i$, Equation (17) can be re-written as

$$\alpha = \frac{0.4 D_{20}{}^4 + 0.2 D_{50}{}^4 + 0.1 D_{65}{}^4 + 0.3 D_{84}{}^4}{D_{50}{}^4} \frac{D_{50}}{D_j} \tag{18}$$

Based on Equation (18), several values of $\alpha$ can be obtained (see Table 3) [2,3,8,27,28].

<p align="center">**Table 3.** Value of $\alpha$ in different equations.</p>

| $D_j=D_{84}$ | $\alpha$ | $D_j=D_{50}$ | $\alpha$ |
|---|---|---|---|
| Barnes (1967) | 2.35 | Barnes (1967) | 5.64 |
| Judd & Peterson (1969) | 2.34 | Bray (1979) | 4.82 |
| Bathurst (1978) | 2.52 | Griffiths (1978) | 4.51 |
| Hey (1979) | 2.51 | | |
| Bathurst (1985) | 2.25 | | |

It is found that $L = 5.50$ and Equations (16) and (17) can better calculate the fixed bed resistance than using the field measured data to verify the fitting curve. Figures 5 and 6 show the comparison and error analysis of the corresponding values in Equation (16), respectively. Moreover, the results indicate that the selection of $\alpha$ is based on the gradation of the bed surface sediment so that there are different $\alpha$ values for different rivers. Equation (16) can basically reduce the error within a better range in calculating the fixed bed resistance of the river channel in the mountainous area.

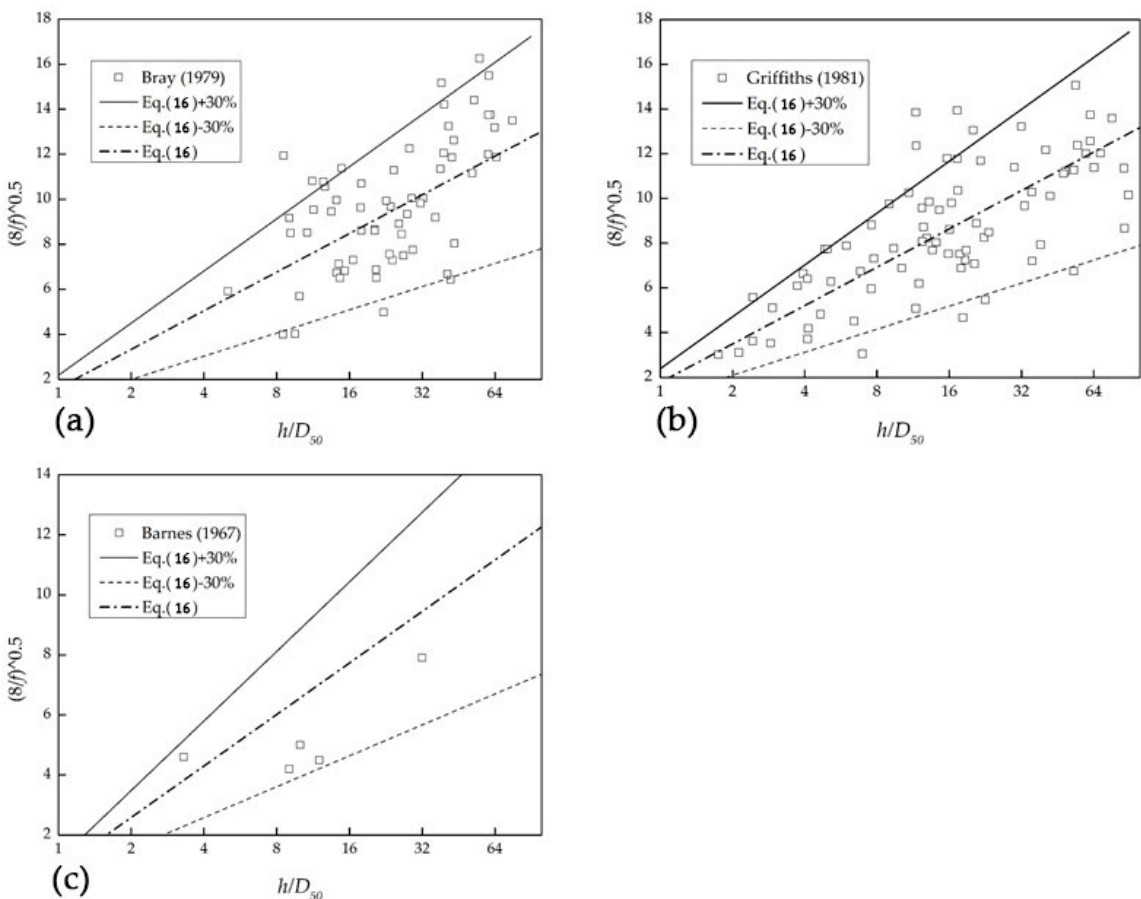

**Figure 5.** The comparison and error analysis of Equation (16) with the measured data (**a**) $D_j = D_{50}, \alpha = 4.82$ , (**b**) $D_j = D_{50}, \alpha = 4.51$, and (**c**) $D_j = D_{50}, \alpha = 5.64$.

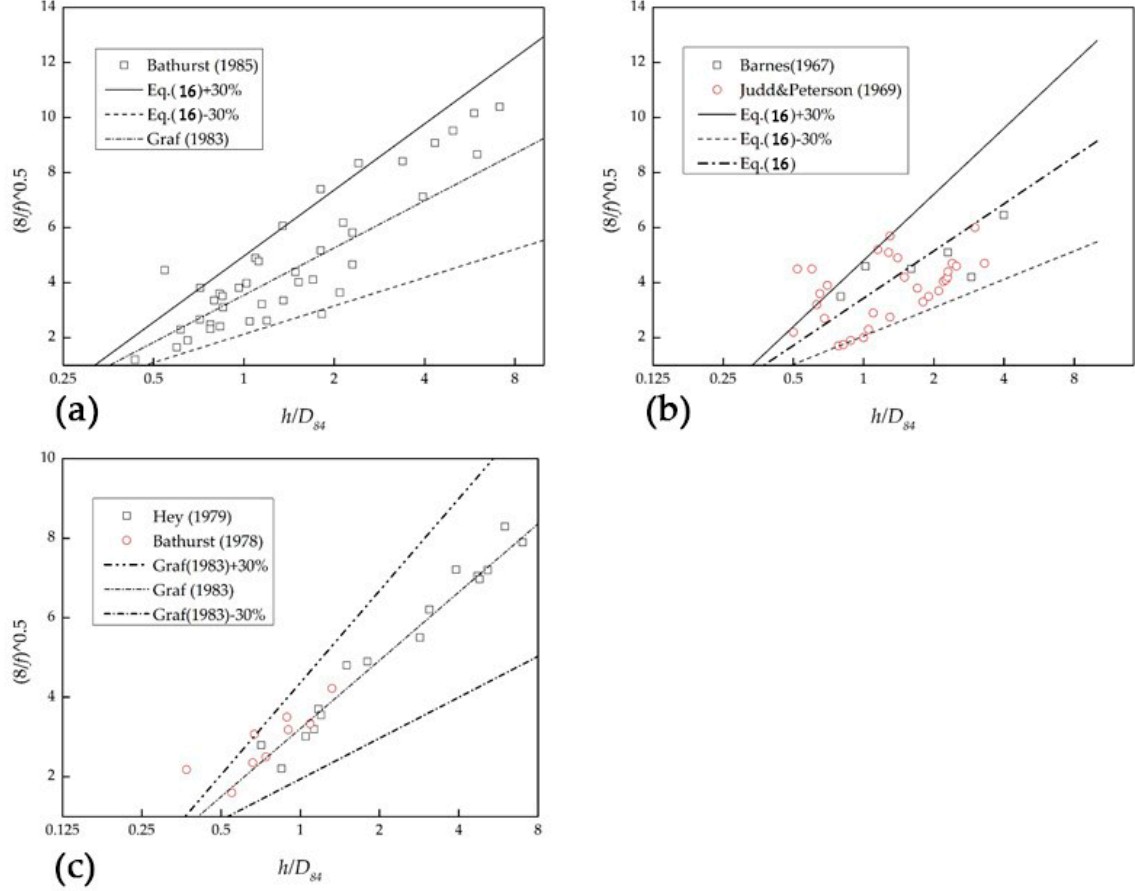

**Figure 6.** The comparison and error analysis of Equation (16) with the measured data (**a**) $D_j = D_{84}, \alpha = 2.25$, (**b**) $D_j = D_{84}, \alpha = 2.35$, and (**c**) $D_j = D_{84}, \alpha = 2.51$

### 3.3. Law of Sediment Laden Flow Resistance in the Steep Gradient Channels

Table 4 shows 58 groups of results with various test conditions.

**Table 4.** Test results for mobile bed resistance flume.

| No. | $Q$ (L/s) | $D_B$ (mm) | $D_T$ (mm) | $h_c$ (cm) | $h$ (cm) | $S$ | $U$ (m/s) | $C_v$ | $f$ | $f_c$ | $f/f_c$ |
|---|---|---|---|---|---|---|---|---|---|---|---|
| 1 | 17.89 | 9.20 | 3.80 | 3.10 | 3.14 | 0.10 | 1.44 | 0.0091 | 0.1181 | 0.1167 | 1.0123 |
| 2 | 17.89 | 9.20 | 3.80 | 3.10 | 3.07 | 0.10 | 1.44 | 0.0302 | 0.1154 | 0.1167 | 0.9890 |
| 3 | 17.89 | 9.20 | 3.80 | 3.10 | 3.44 | 0.10 | 1.44 | 0.0459 | 0.1295 | 0.1167 | 1.0700 |
| 4 | 17.89 | 9.20 | 3.80 | 3.10 | 3.28 | 0.10 | 1.44 | 0.0337 | 0.1235 | 0.1167 | 1.0581 |
| 5 | 24.45 | 9.20 | 3.80 | 3.75 | 3.78 | 0.10 | 1.63 | 0.0167 | 0.1116 | 0.1107 | 1.0080 |
| 6 | 24.45 | 9.20 | 3.80 | 3.75 | 3.88 | 0.10 | 1.63 | 0.0276 | 0.1145 | 0.1107 | 1.0347 |
| 7 | 24.45 | 9.20 | 3.80 | 3.75 | 3.67 | 0.10 | 1.63 | 0.0343 | 0.1083 | 0.1107 | 0.9789 |
| 8 | 35.76 | 9.20 | 3.80 | 4.49 | 4.33 | 0.10 | 1.99 | 0.0309 | 0.0855 | 0.0888 | 0.9635 |
| 9 | 35.76 | 9.20 | 3.80 | 4.49 | 4.52 | 0.10 | 1.99 | 0.0216 | 0.0893 | 0.0888 | 1.0058 |
| 10 | 42.22 | 9.20 | 3.80 | 5.10 | 5.20 | 0.10 | 2.07 | 0.0073 | 0.0952 | 0.0933 | 1.0196 |
| 11 | 40.03 | 9.20 | 3.80 | 5.00 | 4.98 | 0.10 | 2.00 | 0.0267 | 0.0975 | 0.0979 | 0.9960 |
| 12 | 41.67 | 9.20 | 3.80 | 5.08 | 5.03 | 0.10 | 2.05 | 0.0277 | 0.0938 | 0.0947 | 0.9902 |
| 13 | 12.80 | 9.20 | 3.80 | 2.42 | 2.49 | 0.10 | 1.32 | 0.0258 | 0.1116 | 0.1085 | 1.0289 |
| 14 | 12.80 | 9.20 | 3.80 | 2.42 | 2.60 | 0.10 | 1.32 | 0.0370 | 0.1166 | 0.1085 | 1.0744 |

**Table 4.** *Cont.*

| No. | Q (L/s) | $D_B$ (mm) | $D_T$ (mm) | $h_c$ (cm) | h (cm) | S | U (m/s) | $C_v$ | f | $f_c$ | f/$f_c$ |
|---|---|---|---|---|---|---|---|---|---|---|---|
| 15 | 12.80 | 9.20 | 3.80 | 2.42 | 2.78 | 0.10 | 1.32 | 0.0442 | 0.1246 | 0.1085 | 1.0400 |
| 16 | 17.89 | 9.20 | 6.70 | 3.10 | 3.86 | 0.10 | 1.44 | 0.0331 | 0.1453 | 0.1167 | 1.2452 |
| 17 | 17.89 | 9.20 | 6.70 | 3.10 | 3.49 | 0.10 | 1.44 | 0.0307 | 0.1314 | 0.1167 | 1.1258 |
| 18 | 17.89 | 9.20 | 6.70 | 3.10 | 3.31 | 0.10 | 1.44 | 0.0078 | 0.1246 | 0.1167 | 1.1123 |
| 19 | 17.89 | 9.20 | 6.70 | 3.10 | 3.33 | 0.10 | 1.44 | 0.0166 | 0.1254 | 0.1167 | 1.0742 |
| 20 | 24.45 | 9.20 | 6.70 | 3.75 | 4.23 | 0.10 | 1.63 | 0.0234 | 0.1248 | 0.1107 | 1.1280 |
| 21 | 24.45 | 9.20 | 6.70 | 3.75 | 4.40 | 0.10 | 1.63 | 0.0280 | 0.1299 | 0.1107 | 1.1733 |
| 22 | 24.45 | 9.20 | 6.70 | 3.75 | 4.17 | 0.10 | 1.63 | 0.0230 | 0.1231 | 0.1107 | 1.1120 |
| 23 | 35.76 | 9.20 | 6.70 | 4.49 | 4.75 | 0.10 | 1.99 | 0.0179 | 0.0939 | 0.0888 | 1.0579 |
| 24 | 35.76 | 9.20 | 6.70 | 4.49 | 4.85 | 0.10 | 1.99 | 0.0164 | 0.0959 | 0.0888 | 1.0802 |
| 25 | 35.76 | 9.20 | 6.70 | 4.49 | 4.99 | 0.10 | 1.99 | 0.0310 | 0.0987 | 0.0888 | 1.1114 |
| 26 | 41.67 | 9.20 | 6.70 | 5.08 | 5.44 | 0.10 | 2.05 | 0.0184 | 0.1014 | 0.0947 | 1.0987 |
| 27 | 41.67 | 9.20 | 6.70 | 5.08 | 5.68 | 0.10 | 2.05 | 0.0373 | 0.1059 | 0.0947 | 1.2100 |
| 28 | 40.03 | 9.20 | 6.70 | 5.00 | 5.45 | 0.10 | 2.00 | 0.0297 | 0.1067 | 0.0979 | 1.0900 |
| 29 | 12.80 | 9.20 | 6.70 | 2.42 | 3.31 | 0.10 | 1.32 | 0.0493 | 0.1484 | 0.1085 | 1.3678 |
| 30 | 12.80 | 9.20 | 6.70 | 2.42 | 2.74 | 0.10 | 1.32 | 0.0137 | 0.1228 | 0.1085 | 1.1322 |
| 31 | 12.80 | 9.20 | 6.70 | 2.42 | 3.04 | 0.10 | 1.32 | 0.0398 | 0.1363 | 0.1085 | 1.2562 |
| 32 | 12.80 | 9.20 | 6.70 | 2.42 | 2.75 | 0.10 | 1.32 | 0.0168 | 0.1233 | 0.1085 | 1.1100 |
| 33 | 17.89 | 9.20 | 9.20 | 3.10 | 3.91 | 0.10 | 1.44 | 0.0417 | 0.1472 | 0.1167 | 1.3600 |
| 34 | 17.89 | 9.20 | 9.20 | 3.10 | 3.68 | 0.10 | 1.44 | 0.0149 | 0.1385 | 0.1167 | 1.1600 |
| 35 | 17.89 | 9.20 | 9.20 | 3.10 | 3.76 | 0.10 | 1.44 | 0.0242 | 0.1416 | 0.1167 | 1.1800 |
| 36 | 24.45 | 9.20 | 9.20 | 3.75 | 4.32 | 0.10 | 1.63 | 0.0248 | 0.1275 | 0.1107 | 1.1520 |
| 37 | 24.45 | 9.20 | 9.20 | 3.75 | 4.57 | 0.10 | 1.63 | 0.0425 | 0.1349 | 0.1107 | 1.3200 |
| 38 | 35.76 | 9.20 | 9.20 | 4.49 | 5.10 | 0.10 | 1.99 | 0.0283 | 0.1009 | 0.0888 | 1.1359 |
| 39 | 35.76 | 9.20 | 9.20 | 4.49 | 4.94 | 0.10 | 1.99 | 0.0202 | 0.0977 | 0.0888 | 1.1002 |
| 40 | 35.76 | 9.20 | 9.20 | 4.49 | 4.95 | 0.10 | 1.99 | 0.0148 | 0.0979 | 0.0888 | 1.1024 |
| 41 | 49.62 | 9.20 | 9.20 | 6.00 | 6.35 | 0.10 | 2.07 | 0.0147 | 0.1165 | 0.1100 | 1.0980 |
| 42 | 49.04 | 9.20 | 9.20 | 5.90 | 6.50 | 0.10 | 2.08 | 0.0225 | 0.1180 | 0.1071 | 1.1017 |
| 43 | 49.04 | 9.20 | 9.20 | 5.90 | 6.64 | 0.10 | 2.08 | 0.0255 | 0.1206 | 0.1071 | 1.1254 |
| 44 | 12.80 | 9.20 | 9.20 | 2.42 | 3.53 | 0.10 | 1.32 | 0.0375 | 0.1583 | 0.1085 | 1.4587 |
| 45 | 12.80 | 9.20 | 9.20 | 2.42 | 2.94 | 0.10 | 1.32 | 0.0220 | 0.1318 | 0.1085 | 1.2149 |
| 46 | 12.80 | 9.20 | 9.20 | 2.42 | 2.84 | 0.10 | 1.32 | 0.0133 | 0.1273 | 0.1085 | 1.1736 |
| 47 | 49.04 | 9.20 | 12.30 | 5.90 | 6.29 | 0.10 | 2.08 | 0.0044 | 0.1142 | 0.1071 | 1.1600 |
| 48 | 49.04 | 9.20 | 12.30 | 5.90 | 6.21 | 0.10 | 2.08 | 0.0094 | 0.1128 | 0.1071 | 1.1900 |
| 49 | 12.07 | 12.30 | 3.80 | 3.00 | 3.08 | 0.10 | 1.01 | 0.0256 | 0.2387 | 0.2325 | 1.0267 |
| 50 | 12.07 | 12.30 | 3.80 | 3.00 | 3.14 | 0.10 | 1.01 | 0.0470 | 0.2433 | 0.2325 | 1.0467 |
| 51 | 12.07 | 12.30 | 6.70 | 3.00 | 3.47 | 0.10 | 1.01 | 0.0479 | 0.2689 | 0.2325 | 1.1567 |
| 52 | 26.31 | 12.30 | 6.70 | 4.30 | 4.52 | 0.10 | 1.53 | 0.0249 | 0.1515 | 0.1441 | 1.1100 |
| 53 | 34.72 | 12.30 | 6.70 | 5.00 | 5.16 | 0.10 | 1.74 | 0.0293 | 0.1342 | 0.1301 | 1.1230 |
| 54 | 12.07 | 12.30 | 9.20 | 3.00 | 3.75 | 0.10 | 1.01 | 0.0446 | 0.2906 | 0.2325 | 1.2500 |
| 55 | 26.31 | 12.30 | 9.20 | 4.30 | 4.97 | 0.10 | 1.53 | 0.0344 | 0.1665 | 0.1441 | 1.1558 |
| 56 | 35.76 | 12.30 | 9.20 | 5.10 | 5.64 | 0.10 | 1.75 | 0.0239 | 0.1439 | 0.1301 | 1.1059 |
| 57 | 26.31 | 12.30 | 12.30 | 4.30 | 5.00 | 0.10 | 1.53 | 0.0114 | 0.1675 | 0.1441 | 1.1628 |
| 58 | 33.18 | 12.30 | 12.30 | 4.90 | 5.41 | 0.10 | 1.69 | 0.0192 | 0.1480 | 0.1341 | 1.2000 |

Figure 7 compares the results from the fixed bed resistance equation, the ones from the large-scale flume experiments, and the results from the experiment carried out in the present work. It is observed that the sediment movement improves the resistance, so the resistance is larger than that for the fixed bed condition. There are several sets of tests in the experiments with different conditions of the sediment movement. Under this circumstance, the value of $f/f_c$ is less than or equal to 1.0, which indicates that the resistance of the mobile bed does not change in comparison with the fixed bed

resistance. According to the resistance variation of the mobile bed (Equation (1)) [29,30] and Equations (3), (5), and (6), it is found that, when there is a mass movement in the water flow:

$$f_b > 0 \tag{19}$$

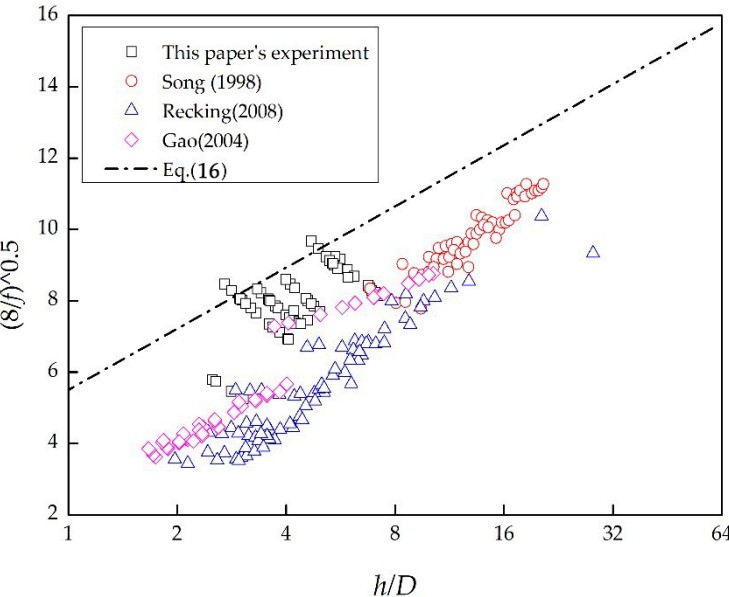

**Figure 7.** Comparison of the results from the fixed bed resistance equation and those from other researchers.

Then:

$$f/f_c > 1 \tag{20}$$

Flow resistance may decrease or remain unchanged, due to the movement of the bed load. Sediments need energy from water to move, when they enter the river, which increases the river resistance. However, for the river channel with a stable bed surface, the coarse particles in the bed load do not exchange with the moving sediment. When the finer sediment enters the channel, it moves in the form of rolling or jumping and fills the coarse grain skeleton of the bed load. With sufficient sediment supplements, the fine sediment covers the original bed surface to form a smoother bed surface. Therefore, although the bed load movement requires a part of the energy to be consumed from the water flow, the resistance decreases, when the fine-grained bed load enters the bed surface with a large bed-grain size. The movement of the fine particles makes the bed surface smoother, resulting in less energy loss of the water flow. If the energy loss is greater than the energy required for the movement of the bed load, then the river resistance may decrease.

However, the resistance may not decrease when small size sediments enter the channel. In the experiment carried out in the present work, when the $D_{50}$ sediment (i.e., particle parameter is 6.7 mm) enters the $D_{50}$ material (i.e., sediment diameter is 9.2 mm and 12.3.mm), the resistance still increases. Therefore, it is speculated that only when the particle size of the sediment carried by the water flow satisfies a certain range may the resistance decrease due to the movement of sediment in the water flow.

It is assumed that the flow velocity in the $y$-direction follows the expression below (see Figure 8):

$$u_y = (1 - c_{sy})u_{cy} + u_{sy}c_{sy} \tag{21}$$

where $u_{cy}$ and $u_{sy}$ are the flow velocity of the clean water and the flow velocity of the sediment in the $y$-direction, respectively. Moreover, $c_{sy}$ is the sediment concentration in the $y$-direction. Performing the integral on Equation (21) yields the average flow velocity as the form below:

$$U = \frac{\int_0^h ((1 - c_{sy})u_{cy} + u_{sy}c_{sy})dy}{h} \tag{22}$$

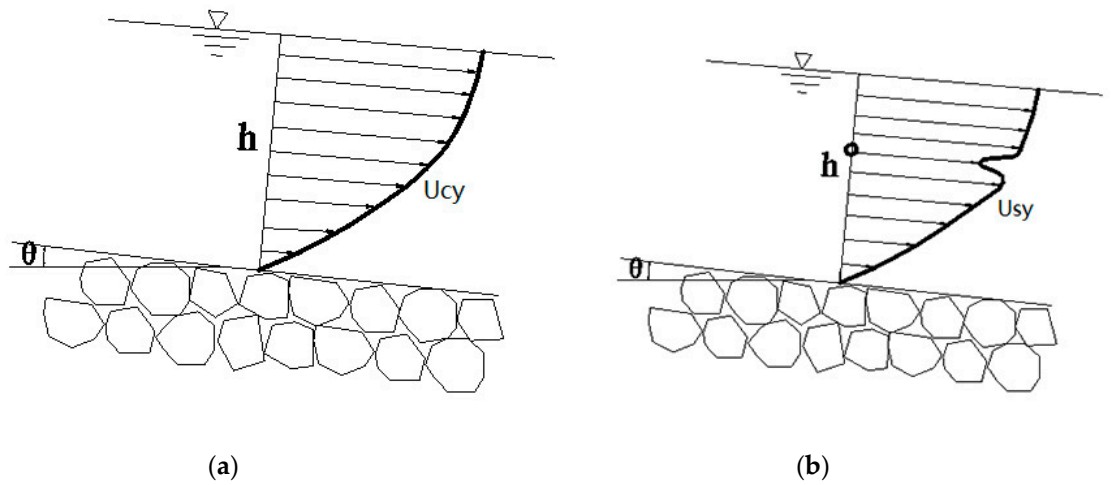

(a)                                             (b)

**Figure 8.** Comparison of the measured and calculated values of sediment particle motion velocity.

Equation (22) is summarized as

$$U = (1 - C_s)\overline{U_c} + U_s C_s \tag{23}$$

where $U_c$ and $U_s$ are the flow average velocity of the clean water and the flow average velocity of the sediment, respectively.

Bagnold [30] found that, when sediment particles move in the water stream, there is a certain difference between the velocity of the sediment particles ($u_s$) and the water velocity ($u_n$) at the position $y_n$. This difference is

$$u_r = u_n - u_s \tag{24}$$

Moreover, the relative velocity between the sediment particles and the water flow movement is equivalent to the sedimentation speed of the sediment particles:

$$u_r = w \tag{25}$$

where $w = 1.72 \sqrt{\left(\frac{\gamma_s - \gamma}{\gamma}\right)gD}$, $Re > 10^3$.

According to Equation (25), it is assumed that there is a certain relationship between the average velocity of sediment particle motion and the sedimentation speed, as the following:

$$U_s = \overline{U_c} - \eta w \tag{26}$$

where $\alpha$ is a constant coefficient.

Combining Equations (26) and (23) yields the following:

$$U = \overline{U_c} - \eta C_s w \tag{27}$$

A series of experiments are carried out on the moving speed of the sediment particles (see Table 5) for the coefficient $\eta$. In the flume with a specific slope of 0.1, the velocity of the sediments with multiple sizes is measured at different water depths. The analysis of the experimental results shows that $\eta$ is a function of the shield stress, in the following form:

$$\eta = \frac{m}{\sqrt{\Theta}} \tag{28}$$

where $m = 0.23$ is a constant coefficient.

$$\Theta = \frac{\rho u_*^2}{(\rho_s - \rho)gD} \tag{29}$$

Figure 9 illustrates a comparison of the velocity of sediment particles, calculated by Equations (26) and (28) with the measured data. The results show that Equations (26) and (28) can accurately predict the velocity of sediment particles.

**Table 5.** Velocity of the particle motion.

| No. | $h$ (m) | $D$ (m) | $J$ | $w$ (u/s) | $U_c$ (u/s) | $U_s$ (u/s) | $u_*$ (u/s) |
|---|---|---|---|---|---|---|---|
| 1 | 0.01 | 0.005 | 0.1 | 0.489 | 1.260 | 1.063 | 0.099 |
| 2 | 0.01 | 0.01 | 0.1 | 0.692 | 1.260 | 0.941 | 0.099 |
| 3 | 0.01 | 0.02 | 0.1 | 0.978 | 1.260 | 0.723 | 0.099 |
| 4 | 0.01 | 0.025 | 0.1 | 1.094 | 1.260 | 0.638 | 0.099 |
| 5 | 0.01 | 0.03 | 0.1 | 1.198 | 1.260 | 0.581 | 0.099 |
| 6 | 0.01 | 0.04 | 0.1 | 1.383 | 1.260 | 0.529 | 0.099 |
| 7 | 0.01 | 0.05 | 0.1 | 1.547 | 1.260 | 0.439 | 0.099 |
| 8 | 0.02 | 0.01 | 0.1 | 0.692 | 2.223 | 2.185 | 0.140 |
| 9 | 0.02 | 0.02 | 0.1 | 0.978 | 2.223 | 1.769 | 0.140 |
| 10 | 0.02 | 0.025 | 0.1 | 1.094 | 2.223 | 1.625 | 0.140 |
| 11 | 0.02 | 0.03 | 0.1 | 1.198 | 2.223 | 1.589 | 0.140 |
| 12 | 0.02 | 0.04 | 0.1 | 1.383 | 2.223 | 1.381 | 0.140 |
| 13 | 0.02 | 0.05 | 0.1 | 1.547 | 2.223 | 1.246 | 0.140 |
| 14 | 0.03 | 0.01 | 0.1 | 0.692 | 2.751 | 2.644 | 0.171 |
| 15 | 0.03 | 0.02 | 0.1 | 0.978 | 2.751 | 2.401 | 0.171 |
| 16 | 0.03 | 0.03 | 0.1 | 1.198 | 2.751 | 2.196 | 0.171 |
| 17 | 0.03 | 0.04 | 0.1 | 1.383 | 2.751 | 2.022 | 0.171 |
| 18 | 0.03 | 0.05 | 0.1 | 1.547 | 2.751 | 1.852 | 0.171 |
| 19 | 0.04 | 0.01 | 0.1 | 0.692 | 2.814 | 2.772 | 0.198 |
| 20 | 0.04 | 0.02 | 0.1 | 0.978 | 2.814 | 2.624 | 0.198 |
| 21 | 0.04 | 0.03 | 0.1 | 1.198 | 2.814 | 2.527 | 0.198 |
| 22 | 0.04 | 0.04 | 0.1 | 1.383 | 2.814 | 2.381 | 0.198 |
| 23 | 0.04 | 0.05 | 0.1 | 1.547 | 2.814 | 2.178 | 0.198 |
| 24 | 0.05 | 0.01 | 0.1 | 0.692 | 2.864 | 2.912 | 0.221 |
| 25 | 0.05 | 0.02 | 0.1 | 0.978 | 2.864 | 2.784 | 0.221 |
| 26 | 0.05 | 0.03 | 0.1 | 1.198 | 2.864 | 2.622 | 0.221 |
| 27 | 0.05 | 0.04 | 0.1 | 1.383 | 2.864 | 2.511 | 0.221 |
| 28 | 0.05 | 0.05 | 0.1 | 1.547 | 2.864 | 2.269 | 0.221 |

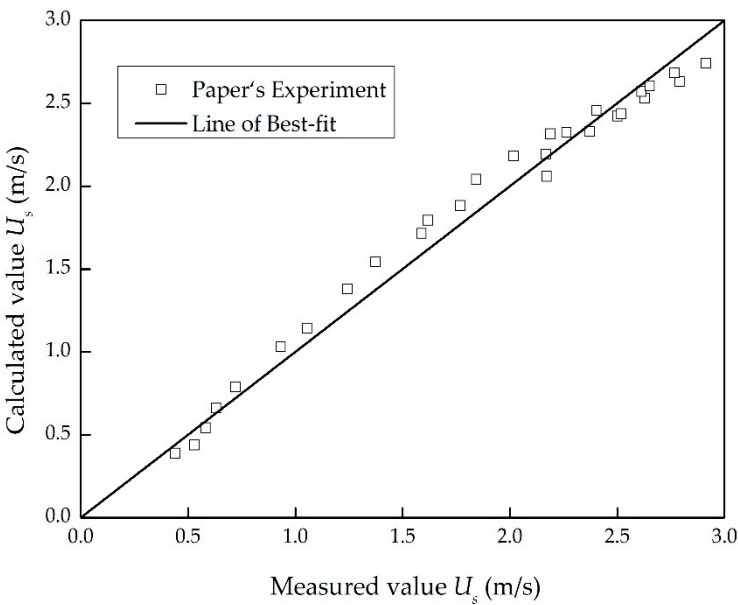

**Figure 9.** Comparison of the measured and calculated values for the sediment particle motion velocity.

The resistance of the sediment laden flow and the clean water can be expressed as

$$\sqrt{\frac{8}{f}} = \frac{U}{U_*} \tag{30}$$

$$\sqrt{\frac{8}{f_c}} = \frac{\overline{U_c}}{U_{*c}} \tag{31}$$

where $U_{*c}$ is the friction velocity in sediment-laden flow.

Combining Equations (30) and (31) yields the following:

$$\frac{f}{f_c} = \left( \frac{\overline{U_c}}{U} \bullet \frac{U_*}{U_{*c}} \right)^2 \tag{32}$$

Considering Equation (27):

$$\frac{U}{\overline{U_c}} = 1 - \frac{\eta C_s w}{\overline{U_c}} \tag{33}$$

$$\frac{U_*}{U_{*c}} = \frac{\sqrt{ghJ}}{\sqrt{gh_c J}} = \sqrt{\frac{h}{h_c}} \tag{34}$$

Applying Equations (33) and (34), Equation (32) can be re-written as

$$\frac{f}{f_c} = \left( \frac{1}{1 - \frac{\eta C_s w}{\overline{U_c}}} \right)^2 \frac{h}{h_c} \tag{35}$$

Resistance and water depth has the following correlation:

$$\frac{h}{h_c} = \left( \frac{f}{f_c} \right)^{1/3} \tag{36}$$

Re-arranging Equations (35) and (36) results in the following equation:

$$\frac{f}{f_c} = \left(\frac{1}{1 - \frac{\eta C_s w}{\overline{U_c}}}\right)^3 \tag{37}$$

The relationship between the sediment transport rate and the ultimate sediment transport rate is uncertain. Therefore, a constant coefficient $\beta$ is added and Equation (37) takes the form below:

$$\frac{f}{f_c} = \left(\frac{1}{1 - \frac{\eta \beta C_s w}{\overline{U_c}}}\right)^3 \tag{38}$$

Therefore, the resistance variation after water flow and sedimentation can be predicted by Equation (38). In order to verify the accuracy of the proposed equation, the results from Equation (38) are compared with data reported by Song et al. [14] and Gao and Abrahams [18] and the results from the experiment carried out in the present study (see Table 6). It is observed that the calculation error is still in the acceptable range and the error value is less than 10%. Figure 10 shows the comparison of the calculation result of Equation (38) and the measured value. It is found that, when $f / f_c$ is small, the calculation result is acceptable. On the other hand, when $f / f_c$ is too large, although the data points are slightly divergent, they are still in the acceptable range.

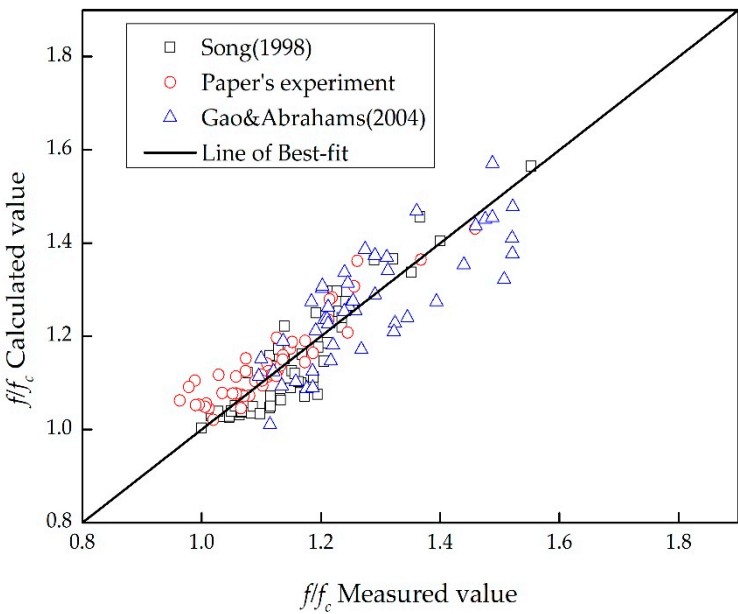

**Figure 10.** Comparison of the results from Equation (38) and those from other works.

**Table 6.** Comparison of the resistance for mobile beds of large gradient channels and those from the Song experiment.

| No. | $h$ (cm) | $S$ (%) | $D$ (mm) | $C_v$ | $f$ | $f_c$ | $f/f_c$ (Exp) | $f/f_c$ (Auth) | Difference (%) |
|---|---|---|---|---|---|---|---|---|---|
| 1 | 20.10 | 0.50 | 12.3 | $3.58 \times 10^{-7}$ | 0.066 | 0.066 | 1.0000 | 1.0054 | 0.54 |
| 2 | 20.90 | 0.50 | 12.3 | $1.15 \times 10^{-6}$ | 0.068 | 0.065 | 1.0462 | 1.0381 | −0.77 |
| 3 | 21.30 | 0.50 | 12.3 | $1.72 \times 10^{-6}$ | 0.066 | 0.064 | 1.0313 | 1.0498 | 1.80 |
| 4 | 21.40 | 0.50 | 12.3 | $1.76 \times 10^{-6}$ | 0.067 | 0.064 | 1.0469 | 1.0506 | 0.36 |
| 5 | 21.80 | 0.50 | 12.3 | $2.37 \times 10^{-6}$ | 0.065 | 0.064 | 1.0156 | 1.0596 | 4.33 |
| 6 | 22.50 | 0.50 | 12.3 | $2.91 \times 10^{-6}$ | 0.067 | 0.063 | 1.0635 | 1.0666 | 0.30 |
| 7 | 22.70 | 0.50 | 12.3 | $3.93 \times 10^{-6}$ | 0.063 | 0.062 | 1.0161 | 1.0754 | 5.83 |
| 8 | 23.50 | 0.50 | 12.3 | $5.74 \times 10^{-6}$ | 0.066 | 0.061 | 1.0820 | 1.0876 | 0.52 |
| 9 | 24.00 | 0.50 | 12.3 | $6.06 \times 10^{-6}$ | 0.065 | 0.061 | 1.0656 | 1.0901 | 2.30 |
| 10 | 24.50 | 0.50 | 12.3 | $5.66 \times 10^{-6}$ | 0.065 | 0.060 | 1.0833 | 1.0890 | 0.53 |
| 11 | 25.00 | 0.50 | 12.3 | $8.19 \times 10^{-6}$ | 0.064 | 0.060 | 1.0667 | 1.1005 | 3.17 |
| 12 | 25.30 | 0.50 | 12.3 | $8.57 \times 10^{-6}$ | 0.063 | 0.060 | 1.0500 | 1.1023 | 4.98 |
| 13 | 16.50 | 0.75 | 12.3 | $7.30 \times 10^{-6}$ | 0.074 | 0.072 | 1.0278 | 1.0656 | 3.68 |
| 14 | 17.10 | 0.75 | 12.3 | $1.57 \times 10^{-5}$ | 0.075 | 0.071 | 1.0563 | 1.0886 | 3.06 |
| 15 | 17.70 | 0.75 | 12.3 | $1.56 \times 10^{-5}$ | 0.076 | 0.070 | 1.0857 | 1.0900 | 0.39 |
| 16 | 18.30 | 0.75 | 12.3 | $3.55 \times 10^{-5}$ | 0.077 | 0.069 | 1.1159 | 1.1151 | −0.08 |
| 17 | 19.10 | 0.75 | 12.3 | $4.17 \times 10^{-5}$ | 0.080 | 0.067 | 1.1940 | 1.1217 | −6.06 |
| 18 | 19.40 | 0.75 | 12.3 | $5.98 \times 10^{-5}$ | 0.077 | 0.067 | 1.1493 | 1.1330 | −1.41 |
| 19 | 19.90 | 0.75 | 12.3 | $7.69 \times 10^{-5}$ | 0.077 | 0.066 | 1.1667 | 1.1416 | −2.15 |
| 20 | 20.30 | 0.75 | 12.3 | $1.19 \times 10^{-4}$ | 0.076 | 0.066 | 1.1515 | 1.1556 | 0.35 |
| 21 | 21.10 | 0.75 | 12.3 | $1.17 \times 10^{-4}$ | 0.074 | 0.064 | 1.1563 | 1.1569 | 0.06 |
| 22 | 13.00 | 0.90 | 12.3 | $3.86 \times 10^{-6}$ | 0.089 | 0.081 | 1.0988 | 1.0322 | −6.05 |
| 23 | 13.60 | 0.90 | 12.3 | $1.08 \times 10^{-5}$ | 0.088 | 0.079 | 1.1139 | 1.0622 | −4.64 |
| 24 | 14.10 | 0.90 | 12.3 | $1.39 \times 10^{-5}$ | 0.087 | 0.078 | 1.1154 | 1.0708 | −4.00 |
| 25 | 14.60 | 0.90 | 12.3 | $2.70 \times 10^{-5}$ | 0.086 | 0.076 | 1.1316 | 1.0910 | −3.59 |
| 26 | 15.30 | 0.90 | 12.3 | $6.00 \times 10^{-5}$ | 0.089 | 0.075 | 1.1867 | 1.1160 | −5.96 |
| 27 | 15.70 | 0.90 | 12.3 | $6.98 \times 10^{-5}$ | 0.086 | 0.074 | 1.1622 | 1.1216 | −3.49 |
| 28 | 16.00 | 0.90 | 12.3 | $9.16 \times 10^{-5}$ | 0.082 | 0.073 | 1.1233 | 1.1304 | 0.63 |
| 29 | 16.50 | 0.90 | 12.3 | $1.50 \times 10^{-4}$ | 0.082 | 0.072 | 1.1389 | 1.1464 | 0.66 |
| 30 | 16.90 | 0.90 | 12.3 | $1.66 \times 10^{-4}$ | 0.080 | 0.071 | 1.1268 | 1.1506 | 2.12 |
| 31 | 17.30 | 0.90 | 12.3 | $1.78 \times 10^{-4}$ | 0.078 | 0.070 | 1.1143 | 1.1538 | 3.55 |
| 32 | 17.80 | 0.90 | 12.3 | $2.10 \times 10^{-4}$ | 0.079 | 0.070 | 1.1286 | 1.1601 | 2.80 |
| 33 | 18.70 | 0.90 | 12.3 | $3.11 \times 10^{-4}$ | 0.084 | 0.068 | 1.2353 | 1.1744 | −4.93 |
| 34 | 12.20 | 1.00 | 12.3 | $2.10 \times 10^{-5}$ | 0.094 | 0.083 | 1.1325 | 1.0725 | −5.30 |
| 35 | 12.80 | 1.00 | 12.3 | $2.90 \times 10^{-5}$ | 0.095 | 0.081 | 1.1728 | 1.0837 | −7.60 |
| 36 | 13.40 | 1.00 | 12.3 | $7.26 \times 10^{-5}$ | 0.095 | 0.080 | 1.1875 | 1.1119 | −6.36 |
| 37 | 13.90 | 1.00 | 12.3 | $1.19 \times 10^{-4}$ | 0.094 | 0.078 | 1.2051 | 1.1280 | −6.40 |
| 38 | 14.30 | 1.00 | 12.3 | $1.48 \times 10^{-4}$ | 0.090 | 0.077 | 1.1688 | 1.1357 | −2.83 |
| 39 | 14.90 | 1.00 | 12.3 | $1.93 \times 10^{-4}$ | 0.092 | 0.076 | 1.2105 | 1.1455 | −5.37 |
| 40 | 15.40 | 1.00 | 12.3 | $3.20 \times 10^{-4}$ | 0.091 | 0.074 | 1.2297 | 1.1623 | −5.49 |
| 41 | 16.20 | 1.00 | 12.3 | $3.50 \times 10^{-4}$ | 0.087 | 0.073 | 1.1918 | 1.1674 | −2.05 |
| 42 | 10.30 | 1.25 | 12.3 | $7.67 \times 10^{-5}$ | 0.098 | 0.091 | 1.0769 | 1.0942 | 1.61 |
| 43 | 11.10 | 1.25 | 12.3 | $1.50 \times 10^{-4}$ | 0.104 | 0.087 | 1.1954 | 1.1169 | −6.57 |
| 44 | 11.70 | 1.25 | 12.3 | $2.54 \times 10^{-4}$ | 0.105 | 0.085 | 1.2353 | 1.1347 | −8.14 |
| 45 | 12.20 | 1.25 | 12.3 | $3.26 \times 10^{-4}$ | 0.103 | 0.083 | 1.2410 | 1.1441 | −7.81 |
| 46 | 12.60 | 1.25 | 12.3 | $3.95 \times 10^{-4}$ | 0.100 | 0.082 | 1.2195 | 1.1513 | −5.59 |
| 47 | 13.80 | 1.25 | 12.3 | $5.83 \times 10^{-4}$ | 0.103 | 0.078 | 1.3205 | 1.2674 | −4.02 |
| 48 | 14.60 | 1.25 | 12.3 | $6.45 \times 10^{-4}$ | 0.098 | 0.076 | 1.2895 | 1.1732 | −9.02 |
| 49 | 15.70 | 1.25 | 12.3 | $6.42 \times 10^{-4}$ | 0.100 | 0.074 | 1.3514 | 1.2765 | −5.54 |
| 50 | 8.40 | 1.50 | 12.3 | $1.73 \times 10^{-4}$ | 0.115 | 0.101 | 1.1386 | 1.1023 | −3.19 |
| 51 | 9.10 | 1.50 | 12.3 | $4.16 \times 10^{-4}$ | 0.119 | 0.097 | 1.2268 | 1.1314 | −7.78 |
| 52 | 9.90 | 1.50 | 12.3 | $5.88 \times 10^{-4}$ | 0.127 | 0.093 | 1.3656 | 1.3455 | −1.47 |
| 53 | 10.50 | 1.50 | 12.3 | $8.05 \times 10^{-4}$ | 0.126 | 0.090 | 1.4000 | 1.3577 | −3.02 |
| 54 | 11.60 | 1.50 | 12.3 | $9.16 \times 10^{-4}$ | 0.132 | 0.085 | 1.5529 | 1.4620 | −5.86 |

## 4. Conclusions

The law of water and sediment movement in the steep gradient channel is a vital basic theory in solving the problem of natural water and material destruction conditions of the mountainous rivers in

Southwest China. Therefore, further in-depth exploration and research on this topic has significant theoretical and practical application value. This paper used the combination of theoretical analysis and experimental research to systematically explore and analyze the fixed and mobile bed resistance of mountainous rivers. The following conclusions have been obtained:

1. Based on the existing results, a calculation scheme is established to simulate the velocity distribution of the uniform sediment bed on the steep gradient channels. Based on the test data, the proposed expressions are verified to have a reasonable performance;
2. Through the existing research results, an expression is determined to calculate the resistance of the fixed bed with non-uniform sediment. Moreover, the selection of the roughness size is analyzed to put forward a new calculation method. The expression for the resistance of the non-uniform bed surface of the steep gradient channels is obtained through analyzing the measured field data;
3. When calculating the resistance of the mobile bed in mountainous rivers, the fixed bed resistance cannot be used. Experimental results show that, when the bed surface is stable and the bed load particle size is much smaller than that of the gravel, the bed surface is smooth and flat. At this point, there are several possibilities so the resistance may increase, may remain unchanged, or even may decrease;
4. By assuming the flow velocity distribution of clean water and sediment laden flows, the variation law for the resistance of sediment laden flow is derived. Moreover, the relationship between the velocity of the sediment movement and the average flow velocity is obtained. It is found that the proposed method can accurately predict the variation of the sediment carrying resistance.

Because of the various influencing factors and complicated water characteristics in the mountainous rivers, the research in this field is still not mature enough. There are still a lot of possibilities to be explored in the study of river resistance and sediment transport characteristics in steep gradient mountainous areas.

**Author Contributions:** Conceptualization, J.H.; Investigation, J.H. and Q.Z.; Methodology, C.Z. And J.H.; Validation, J.H and F.L.; Visualization, J.H.; Writing—original draft, J.H. and Z.Y.; Writing—review and editing, J.H. and D.W.

**Funding:** This project is supported by the Chongqing Research Program of Application Foundation and Advanced Technology [grant number cstc2016jcyjA1937/1935], science and technology project of Chongqing Education Committee [grant number KJ1600535/KJ1600514], and Open Fund Project of Key Laboratory for Inland Waterway Regulation Engineering of Ministry of communication in Chongqing Jiaotong University [grant number NHHD-201513/201505].

**Acknowledgments:** Many thanks to the Key Laboratory for Inland Waterway Regulation Engineering of Ministry of communication in Chongqing Jiaotong University.

**Conflicts of Interest:** The authors declare no conflict of interest. The founding sponsors had no role in the design of the study; in the collection, analyses, or interpretation of data; in the writing of the manuscript, and in the decision to publish the results.

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
