# Peer review of "Fixed-Bed and Mobile-Bed Resistance of Channels with Steep Gradients in Mountainous Areas"

_water, doi:10.3390/w11040681_

Round 1
Reviewer 1 Report
The manuscript presents results of experimental research on the resistance of bed load materials in steep, mountain channels. Unfortunately, I had great difficulty trying to understand its content. There are extensive grammar, sentence structure, and terminology issues with this manuscript. The paper is written in a way that detracts from the clarity of its content. The manuscript needs extensive revision and copyedits before I can properly review the work that is presented. For now, I am answering "Not applicable" in the review categories above until the writing is significantly improved.
Author Response
We have rewritten some parts of the paper and corrected some grammar mistakes such as “Some “s” are missing at the end of verbs”. We also reduced some terminology issues such as replacing “sink” by “flume”.
Reviewer 2 Report
The paper presents experiments and method to evaluate the flow resistance with fixed and mobile –bed.
A large number of significant experiments were carried out and compared with previous Scholar data. The advantage of taking into account the bed movement for resistance is well-presented. However I have some comments that can help this paper to be publishable.
1 –Variables are not clearly defined : u1, Uc, Us, U*c. Or some confusions are possible: C is the chezy coefficient and the mass concentration, A is used 2 times (eq. 3.8 and 3.20) alpha is used in Eq. 3.9 and 3.20. Precise the unit of C.
2- Previous studies on steep slope and roughness are not quoted ( Rice 1998, Pagliara et al. 2006, Ferguson 2007, Cassan et al. 2017).
Ferguson, R. (2007), Flow resistance equations for gravel- and boulder-bed streams, Water Resour. Res., 43, W05427, doi:10.1029/2006WR005422.
Cassan L, Roux H, Garambois P-A. A Semi-Analytical Model for the Hydraulic Resistance Due to Macro-Roughnesses of Varying Shapes and Densities. Water, MDPI, 2017, 9 (9),pp.637.
J. Aberle & G. M. Smart (2003) The influence of roughness structure on flow resistance on steep slopes, Journal of Hydraulic Research, 41:3, 259-269, DOI: 10.1080/00221680309499971
3-Experiments setup part has to be more detailed: what is the slope, how the sediment velocity is obtained, i.e is the sediment rate at the end of the flume representative during experiments ?
4- What is the formula applied for w?
5- legend sof Figs. 3.6 and 3.8 must be corrected. Could you explain why the comparison is eq.3.8 or with eq. 3.6.
6-I propose to merge figs 3.3 to 3.5 and 3.6 to 3.8 to enhance comparison of several data, the sensitivity to alpha would be shown.
7- Eqs. 1.4 and 3.15 must be corrected.
8- Some “s” are missing at the end of verbs.
9- What is the B value when you determine z0=0.2 D. Could you illustrate the calibration method?
10- for eq.3.2, could you precise what is the reference for vertical coordinate (bed , top of roughness element)? It could be important for large particle size.
11- I would replace “sink” by “flume”.
12- L285-287 are not clear.
13- L308-310 are not clear.
14- Eq 3.13 and 3.18, could you give physical considerations for your assumptions? (Stockes number, steep slope, energy balance,…)
15-Eq 3.30 is clearly deduced from the hypothesis on the ratio w/Uc. Then comment on this point, could help the discussion on the range of validity.
16 –references L432, 438, 440 have to be rewritten.
17-Typo L33 (+),67 (expression of expression), 391 (, At).
18- Table 3 is not in the text.
Author Response
Point 1: Variables are not clearly defined : u1, Uc, Us, U*c. Or some confusions are possible: C is the chezy coefficient and the mass concentration, A is used 2 times (eq. 3.8 and 3.20) alpha is used in Eq. 3.9 and 3.20. Precise the unit of C.
Response 1:
1. u1 has been replaced as u0. A of 3.20 has been replaced as m. Alpha has been replaced as
2. We have updated the definition:
Where: and are respectively the flow average velocity of the clean water and the flow average velocity of the sediment. Where: is friction velocity in sediment-laden flow.
3. The C to the mass concentration has changed to Cv by us.
4. The unit of C is L^(1/2)T^(-1)
Point 2: Previous studies on steep slope and roughness are not quoted ( Rice 1998, Pagliara et al. 2006, Ferguson 2007, Cassan et al. 2017).
Ferguson, R. (2007), Flow resistance equations for gravel- and boulder-bed streams, Water Resour. Res., 43, W05427, doi:10.1029/2006WR005422.
Cassan L, Roux H, Garambois P-A. A Semi-Analytical Model for the Hydraulic Resistance Due to Macro-Roughnesses of Varying Shapes and Densities. Water, MDPI, 2017, 9 (9),pp.637.
J. Aberle & G. M. Smart (2003) The influence of roughness structure on flow resistance on steep slopes, Journal of Hydraulic Research, 41:3, 259-269, DOI: 10.1080/00221680309499971
Response 2: All the papers that you mentioned has been quoted. We have rewritten the reference.
Point 3: Experiments setup part has to be more detailed: what is the slope, how the sediment velocity is obtained, i.e is the sediment rate at the end of the flume representative during experiments ?
Response 3:
1. The slope of this experiment is 10%.
2. A funnel is arranged in the inlet section, we can control the sediment supplement rate through the opening of the funnel, and devices is installed at the end of the flume to measure the sediment transport rate.
3. All the details can been seen in this paper.
Point 4: What is the formula applied for w?
Response 4:
1. , . All related information has been added to the paper.
Point 5: legend sof Figs. 3.6 and 3.8 must be corrected. Could you explain why the comparison is eq.3.8 or with eq. 3.6.
Response 5:
1. The Figs of this paper has been corrected.
2. The reason is that: Using the field measured data to verify the fit, it is found that formula (3.8) can calculate the fixed bed resistance better. Figure 3.3 - Figure 3.8 show the comparison and error analysis of the corresponding values in the formula (3.8), respectively. The results show that the selection of α is based on the gradation of the bed surface sediment, for different rivers, the value is different, and the formula (3.8) can basically control the error in a better range in calculating the fixed bed resistance of the river channel in the mountainous area.
Point 6: I propose to merge figs 3.3 to 3.5 and 3.6 to 3.8 to enhance comparison of several data, the sensitivity to alpha would be shown.
Response 6:
The Figs of this paper has been corrected.
Point 7: Eqs. 1.4 and 3.15 must be corrected.
Response 7:
The Eqs of this paper has been corrected.
Point 8: Some “s” are missing at the end of verbs.
Response 8:
The grammar mistakes has been corrected.
Point 9: What is the B value when you determine z0=0.2D. Could you illustrate the calibration method?
Response 9 :
We first obtain the value of Z0 according to the data of Abigrre-Pe et al. (1990) combined with Bathurst (1977), Thompson and Campbell (1979) and Griffiths (1981), etc., at this time, the value of B was undetermined . Then we fit line according to the experimental data in this paper, and then compares it with Song (1994), Recking (2008) and the formula of this paper to obtain the B value.
Point 10: for eq.3.2, could you precise what is the reference for vertical coordinate (bed , top of roughness element)? It could be important for large particle size.
Response 10:
This part has been corrected as:
Aguirre-Pe et al. and Graf et al. have pointed out that the flow velocity above the surface of the bed which is reference for vertical coordinate is basically in line with the logarithmic distribution, as follows
Point 11: I would replace “sink” by “flume”
Response 11:
“sink” has been replaced by “flume”
Point 12: L285-287 are not clear.
Response 12:
This part has been corrected as:
At this time, the value of is less than or equal to 1, which means the resistance of mobile bed does not change (even decreases) in comparison with fixed bed resistance.
Point 13: L308-310 are not clear.
Response 13:
This part has been corrected as:
However, it is not the resistance of the channel that is reduced as long as the particles with smaller particle size of the bed sand enter the channel. In the test, when the sediment with as 6.7 mm entered the sand with as 9.2 mm and 12.3.mm, the resistance still increased. Therefore, it is speculated that only when the particle size of the sediment carried by the water flow satisfies a certain range, the resistance may decrease due to the movement of sediment in the water flow.
Point 14: Eq 3.13 and 3.18, could you give physical considerations for your assumptions? (Stockes number, steep slope, energy balance,…)
Response 14:
In the article, the speed should be related to Stockes number or slope, and we use the flow rate of fresh water and the flow rate of sediment to describe the new flow rate relationship. The new relationship is only related to the assumed speed. Therefore, no physical considerations are given here.
Point 15: Eq 3.30 is clearly deduced from the hypothesis on the ratio w/Uc. Then comment on this point, could help the discussion on the range of validity.
Response 15:
The article focuses on the verification using Song et al. (1998), Gao and Abrahams (2004) and the experimental data in this paper. The error of Equation 3.8 was found to be small. This implies a range. However, there is no in-depth study of the larger scope of this validity. Based on your inspiration, We will focus on whether the study has a more general formula in future research.
Point 16: references L432, 438, 440 have to be rewritten.
Response 16:
The references have been rewritten.
Point 17: Typo L33 (+),67 (expression of expression), 391 (, At).
Response 17:
The mistakes have been corrected.
Point 18: Table 3 is not in the text.
Response 18:
The table has been added to the text. “Table 3 Value ofin Different Equation”
Round 2
Reviewer 1 Report
A persistent issue between the two versions of the manuscript I
have reviewed is use of the English language/style. The paper is still written in a way that detracts from
the clarity of its content.There are significant English-language errors that need to be corrected, including proper word usage
and sentence construction. I started to correct some of the
grammar errors, but there are far too many and I essentially ended up
rewriting the abstract as a result (as an example). I strongly suggest you
find someone fluent in English who can copyedit and significantly rewrite the manuscript. I cannot approve acceptance of the manuscript until this issue is fully addressed.
Title: Fixed-bed and mobile-bed resistance of channels with steep gradients in mountainous areas
Abstract: Flood discharge and sediment transport are closely linked to channel resistance in steep mountain streams. Previous research has focused mainly on the resistance of fixed-bed channels with steep gradients and mobile-bed channels in alluvial rivers. The experimental research presented here establishes a calculation method for fixed-bed resistance of mountain channels to complement this earlier work. The basic expression of the mobile-bed resistance of steep mountain channels is derived by determining the controlling factors of bed load movement on the riverbed resistance. The proposed formula can predict the variation of the bed load resistance. The results of this research will improve understanding of fluid dynamics and sediment transport in steep mountain channels.
Line 28: “rivers” instead of “river”
Line 29: “valleys are narrow and the slopes are steep.”
Line 30: What are steps-deep-tank terrain?
Line 30: “bed sand is comprised mostly of pebbles and gravel”
Line 35-36: Calculation of what sediment? Be specific.
Line 37: What do you mean by the sediment not having been started?
Line 49 and 51: You mention the differences twice and even state they cannot be ignored, but do not mention what they are.
Line 93: Who is “they”?
Line 105: Identify the researchers as you did with Gao and Abrahams in line 99. This would be helpful throughout the Introduction to show the transition between various supporting studies.
Lines 123-124: Why did you use a slope of 10%? Does this relate well to the average gradients in mountain streams of southwest China?
Lines 138-140: Pebbles are the base layer and sand is added on top? How large were the pebbles? What factors were considered when selecting particle size of the bed sand?
Lines 142-143: Why did you choose these 4 particle sizes?
Experimental setup: How many times were the experiments run?
Lines 209-213 should serve as an example for the type of clarity in writing that is needed throughout the manuscript.
Line 221-222: Source for this assertion?
Line 409: Can you provide examples of some of these research possibilities?
Author Response
Response to Reviewer 1 Comments
Point 1: Line 28: “rivers” instead of “river”
Response 1: We have corrected this mistake.
Point 2:Line 29: “valleys are narrow and the slopes are steep.”
Response 2: We have corrected this mistake.
Point 3: Line 30: What are steps-deep-tank terrain?
Response 3 We have deleted the phase “steps-deep-tank terrain”, which describes the transition area between the end of slope and a tank incorrectly.
Point 4: Line 30: “bed sand is comprised mostly of pebbles and gravel”
Response 4: We have corrected this mistake.
Point 5: Line 35-36: Calculation of what sediment? Be specific.
Response 5: “Sediment” refer to “uniform and non-uniform bed material”, we have rephrase this sentence.
Point 6: Line 37: What do you mean by the sediment not having been started?
Response 6: We have rephrased the sentence as “When the incipient starting phenomenon does not exist in the mountainous river or there is low-intensity sediment transport, we regard the influence of the presence of sediment particles on the bed surface as the fixed bed resistance of steep gradient channel.”
Point 7: Line 49 and 51: You mention the differences twice and even state they cannot be ignored, but do not mention what they are.
Response 7:
The sentences have been rewritten as “Previous flume experiment or field data showed that the resistance of the steep gradient channel and alluvial rivers are related to the size, shape, gradation of sediment particles and the geometric shape of rivers. The characteristics between them are different. The different results between quantitative studies of the resistance of mountainous and alluvial rivers cannot be ignored.”
Point 8: Line 93: Who is “they”?
Response 8:
“They” refers to “Gao and Abrahams (2004) ”, which has been rewritten.
Point 9: Line 105: Identify the researchers as you did with Gao and Abrahams in line 99. This would be helpful throughout the Introduction to show the transition between various supporting studies.
Response 9 :
We have identified the researchers in line 95.
Point 10:Lines 123-124: Why did you use a slope of 10%? Does this relate well to the average gradients in mountain streams of southwest China?
Response 10:
The slope here related well to the mean gradients in mountainous rivers of southwest China.
Point 11: Lines 138-140: Pebbles are the base layer and sand is added on top? How large were the pebbles? What factors were considered when selecting particle size of the bed sand?
Response 11:
Pebbles are the base layer and sand is added on top, the average diameter of peddles is 30mm. We have considered our field data, particle sizes of previous studies and the actual maximum discharge the channel can provide with.
Point 12:Lines 142-143: Why did you choose these 4 particle sizes?
Response 12:
Focusing on the reliability, representative and comparability of this paper’s results, we mainly considered the value of sizes of previous studies that the paper has mentioned combined with our filed data has done before.
Point 13:Experimental setup: How many times were the experiments run?
Response 13:
Basically, the total number of trials could be more than 100 times. We have around 76 valid tests.
Point 14:Lines 209-213 should serve as an example for the type of clarity in writing that is needed throughout the manuscript.
Response 14:
We have changed the content as an explanation.
Point 15:Line 221-222: Source for this assertion?
Response 15:
Related references has been added to support the assertion.
Point 16:Line 409: Can you provide examples of some of these research possibilities?
Response 16:
The law of water and sediment movement in the steep gradient channel is a vital basic theory in solving the problem of natural water and sand conditions destruction of the mountainous rivers in Southwest China. Further in-depth exploration and research has significant theoretical value and practical application value. For instance, in order to know the flood discharge and sediment carrying capacity of the river, it is necessary to understand the flow and sediment movement in the channel. In a certain riverbed shape and slope, the discharge volume, the flow through the main channel, the overbank flow, and the maximum flow velocity are closely related to the resistance of the channel to the flow. Moreover, the river resistance reflects the influence of the action of water flow on the riverbed, and also determines the movement intensity of sediment.
Reviewer 2 Report
The authors improved their paper and I think it is suitable for publications. However I have some additional comments :
L 103 : is S defined more precisely before?
Figure 3.1 : is “wake “region is the adequate word? This zone is above roughness and there is a confusion with the law of the wake in a turbulent layer (near the surface).
Figure 3.7 is not very explicit. I think the velocity ucy and usy have to be indicated in the figure/
L 349 : What is the m value?
Author Response
Response to Reviewer 2 Comments
Point1: L 103 : is S defined more precisely before?
Response 1: Specifically, S is defined as the gradient of Water surface. The sentence is rephrased.
Point 2:Figure 3.1 : is “wake “region is the adequate word? This zone is above roughness and there is a confusion with the law of the wake in a turbulent layer (near the surface).
Response 2:The word has been removed. And the picture has been repainted.
Point 3: Figure 3.7 is not very explicit. I think the velocity ucy and usy have to be indicated in the figure/
Response 3: The Figure has been repainted.
Point 4: L 349 : What is the m value?
Response 4: M=0.23, which has been added to the content.
Round 3
Reviewer 1 Report
Thank you for addressing many of my comments and concerns. The science of this manuscript is sound. Once again, my main issue is the use of the English language. The grammar issues of the manuscript still detract from the science and make understanding the manuscript difficult.
Author Response
We have polished our language and fixed any other issues about grammar in the revised manuscript.
This manuscript is a resubmission of an earlier submission. The following is a list of the peer review reports and author responses from that submission.